# Validation of a Novel ELISA for the Diagnosis of Hemorrhagic Septicemia in Dairy Cattle from Thailand Using a Bayesian Approach

**DOI:** 10.3390/vetsci7040163

**Published:** 2020-10-28

**Authors:** Tawatchai Singhla, Pallop Tankaew, Nattawooti Sthitmatee

**Affiliations:** 1Department of Food Animal Clinic, Faculty of Veterinary Medicine, Chiang Mai University, Chiang Mai 50100, Thailand; 2Central laboratory, Faculty of Veterinary Medicine Chiang Mai University, Chiang Mai 50100, Thailand; pallop.tank@cmu.ac.th; 3Department of Veterinary Bioscience and Veterinary Public Health, Faculty of Veterinary Medicine, Chiang Mai University, Chiang Mai 50100, Thailand; nattawooti.s@cmu.ac.th

**Keywords:** bayesian latent class analysis, hemorrhagic septicemia, indirect ELISA, *Pasteurella multocida*

## Abstract

The objective of this study was to estimate sensitivity (Se) and specificity (Sp) of a novel enzyme-linked immunosorbent assay (ELISA) test (using a coating antigen from *Pasteurella multocida* M-1404 via heat extraction) and an indirect hemagglutination (IHA) test for detection of Hemorrhagic septicemia (HS) in dairy cows, under Thai conditions, using a Bayesian approach. Dairy cow sera with a total of 1236 samples from 44 farms were tested with the two tests to detect immune responses against the HS. Percentages of positive samples for the ELISA and IHA tests were 73% (901/1236) and 70% (860/1236), respectively. Estimated sensitivity and estimated specificity of the ELISA test were 90.5% (95% posterior probability interval (PPI) = 83.2–95.4%) and 70.8% (95% PPI = 60.8–79.8%), respectively. Additionally, estimates for the Se and Sp values of the IHA test were 77.0% (95% PPI = 70.8–84.1%) and 51.1% (PPI = 36.8–66.3%), respectively. The estimated prevalence of the disease was 71.7% (95% PPI = 62.7–82.6%). These results demonstrate that the ELISA test can be a useful tool for the detection of the presence of an antibody against the HS in dairy cows. Notably, the cows in this area indicated a high percentage of exposure to *Pasteurella multocida*.

## 1. Introduction

Hemorrhagic septicemia is an acute, fatal and septicemic disease of buffaloes, beef, and dairy cows. The disease is mainly caused by *Pasteurella multocida* (*P. multocida*), which consists of specific serotypes, including B:2 (Asian serotype) and E:2 (African serotype). The pathogen is a gram-negative non-endospore forming bacteria and is known as a multiple killing agent [1]. Case fatality rates of relevant disease outbreaks may be above 80% [2]. It has been reported that mortality losses (direct losses) account for 92.7% of total losses, while the rest (7.3%) were due to treatment costs, growth reduction, and low milk yields [3]. Recently, research in Thailand using an indirect enzyme-linked immunosorbent assay (ELISA) has reported that the seroprevalence of the disease in dairy cows was 35.7% [4].

Normally, the indirect hemagglutination (IHA) test has been used to detect a specific antibody against *P. multocida*, but it is not typically used for diagnosis [5]. Animals with high IHA titers indicate recent contact with the bacteria [5]. However, a study of IHA test performance is scarce, and a previous study and an expert opinion have suggested that the sensitivity (S) and specificity (Sp) are imperfect [4]. Therefore, this method cannot be used as a reference test for the evaluation of novel test performance. Recently, Thai researchers have developed an indirect ELISA test to detect specific antibodies against *P. multocida* in dairy cows using a coating antigen from *P. multocida* M-1404 (B:2) via heat extraction [4]. The researchers have validated the ELISA test in positive and negative control sera and suggested that this method has displayed reliable degrees of performance with Se and Sp values of 92.1 and 71.9, respectively. However, the ELISA and IHA tests have never been proven their performance of this antibody detection on large sample sizes and multiple areas in Thailand.

Determination of the sensitivity and specificity of a screening test is usually done by comparing it to a reference test (a gold standard) or applying to known infected animals and known healthy animals. The sensitivity and specificity can be estimated directly if a gold standard is available [6] However, the perfect reference test and a large population of known infected animals are unavailable. These are the major problems for the estimation of diagnostic test performance in this study. Moreover, serious bias may be introduced in the evaluation of the accuracies of the novel test when classification errors in the reference test are overlooked [7,8]. Thus, a Bayesian latent class analysis was performed to address these problems because this method has inferred posterior estimates from a combination of prior information and observed data to solve the uncertainty of unknown parameters such as test performance or disease prevalence [9]. This analysis has become increasingly used in veterinary epidemiology to evaluate the accuracy of a diagnostic test or the true disease prevalence when a gold standard may be impossible to obtain [9,10,11,12].

The objective of this study was to estimate the performance of a novel indirect ELISA (using coating antigen from *P. multocida* M-1404 (B:2) via heat extraction) and IHA tests among dairy cows in Thailand, using a Bayesian approach.

## 2. Materials and Methods

### 2.1. Blood Samples

The dairy cow sera (*n* = 1236) from 44 farms located in Sanpatong, Mae-Wang, and Doi-Loh districts, Chiang Mai province, were obtained from the Chiang Mai Provincial Livestock Office, the Department of Livestock Development (DLD), Thailand. Serum samples were collected from non-vaccinated dairy cows (age ≥ 1 year) by the DLD annual disease investigation program regardless of health and production status.

### 2.2. Enzyme-Linked Immunosorbent Assay (ELISA) Test

Before the testing, positive serum controls were prepared by immunization of healthy dairy cows with a formaldehyde-fixed *P. multocida*, as described previously [13]. In addition, colostrum-deprived neonatal calf sera were used as a negative serum control. The dairy cow sera were tested by an indirect ELISA test for detection of the specific antibody against *P. multocida*, following the protocol as described elsewhere [4]. Briefly, the indirect ELISA test was performed using the *P. multocida* M-1404 (B:2) heat extracted antigen as a coating antigen. Each well was coated with 100 µL of the diluted coating antigen. The plates were incubated at 4 °C overnight. After three washes with washing buffer (PBST; 0.05% Tween 20 in 0.001 M PBS, pH 7.4), each well was blocked with 100 µL of blocking buffer at 37 °C for 1 h. Then, washed the plates thrice, and the wells were added with 100 µL of dairy cow serum diluted in blocking buffer. Following incubation at 37 °C for 1 h, the plates were washed three times, and the wells were added with 100 µL of HRP-conjugated goat anti-bovine IgG (KPL) diluted in blocking buffer. The plates were then incubated at 37 °C for 1 h. Following another three washes, the wells were added with 100 µL of tetramethylbenzidine (KPL) diluted in substrate buffer, and the plates were incubated in the dark at 37 °C for 30 min. Then, the color reaction was stopped by adding 50 μL of 3 M H_2_SO_4_. The results were presented using optical density (OD) using an automatic ELISA plate reader (AccuReader; Metertech, Taipei, Taiwan) for reading absorbance at a wavelength of 450 nm. The cut-off value was 0.128, which was calculated from the mean OD of negative control sera, plus three standard deviations, as described in the previous study [4]. A sample was considered as a positive sample when both (1) the difference between the mean OD of the sample and with PBS alone, and (2) the difference between the mean OD of the sample and negative control ODs were greater than the cut-off value [4].

### 2.3. Indirect Hemagglutination Assay (IHA) Test

The dairy cow sera were also tested by an IHA test to classify negative and positive samples according to the guideline [5]. Briefly, a microtiter system was used to perform the IHA test. Serial two-fold dilutions of antiserum were produced in BSA-PBS, followed by adding 0.025 mL of the sensitized sheep red blood cells (SRBC) to 0.025 mL of the antiserum dilution in U-bottom plates. Following the shaking step, the plates were allowed to stand at 25 °C for 1–2 h for reading SRBC settling patterns. The IHA titer was presented as the reciprocal of the highest dilution of serum, showing a definite positive pattern (flat sediment) compared to the negative control pattern (smooth dot in the center of the well). Unsensitized SRBC plus test serum and sensitized SRBC plus diluent were used as controls. Before being used in the IHA test, heterophile antibodies were removed by absorption with unsensitized GA-SRBC at 25 °C for 2 h when they were detected in the sera [13]. The samples were considered positive at IHA test if titers from 1:160 up to 1:1280 or higher were detected [5].

### 2.4. Sensitivity and Specificity Estimation

A Bayesian latent class modeling was conducted to estimate Se and Sp values of the indirect-ELISA and IHA tests, as described in the previous reports [9,14]. The reaction mechanism of the two tests was based on the specific antibody detection; therefore, the results of the tests were defined as conditional dependence upon each other [14]. The sera were assumed that they were from the same population because these samples were obtained from farms located in the same area, as suggested in previous studies [10,15]. Therefore, a Bayesian model for two conditionally dependent tests and one population was adjusted to the observed data for estimation of the Se and Sp values of the two tests and the true disease prevalence.

The Bayesian latent class model defined that for the *k* populations, the counts (Y*_k_*) of the different combinations of test results such as +/+, +/−, −/+, or −/− for the two tests followed a multinomial distribution: Y*_k_* | P*_qrk_* ~ multinomial (n*_k_*, {P*_qrk_*}), where *qr* was the multinomial cell probability for the two-test outcome combination, and P*_qrk_* was a vector of the probabilities of observing the individual combinations of the test results. The prior values of all parameters (Se, Sp, and prevalence), shown in Table 1, were introduced to the model using beta distribution based on the findings of a previous report [4]. The means of the central values provided in this study were chosen as being the most likely values, while the lowest modal value was defined as a 97.5% lower limit for prior distributions (i.e., above the lower limit of the 95% interval for the prior). After analysis, the median of the posterior distributions was used for point estimates. The standard deviation of a marginal posterior distribution referred to a 95% posterior probability interval (PPI) [4,11]. All analyses were conducted in JAGS 3.4.0 via the rjags and R2jags packages, using R v3.2.2 [16,17,18,19]. Posterior distributions were based on 100,000 iterations after the burn-in phase of 10,000 iterations were discarded.

The Gelman-Rubin diagnostic plots were inspected for model convergence and a lack of autocorrelation assessment [20,21]. The model convergence was identified when the upper limit was close to one. The model was tested for the goodness of fit using the Deviance Information Criterion (DIC) and the number of effectively estimated parameters (pD) as calibrating parameters [22,23]. A sensitivity analysis was conducted by replacing each prior value with a non-informative uniform 0–1 distribution to evaluate the influence of the prior distributions and the conditionally dependent assumption between the ELISA and IHA tests on the posterior distributions [9].

As an OIE-recommended assay for the detection of antibodies against *P. multocida*, the IHA test was also used as the gold standard to estimate the Se and Sp of the ELISA test by classical epidemiological calculation or direct comparison.

## 3. Results

### 3.1. Results from Diagnostic Tests

The diagnostic test results are shown in Table 2. Of 1236 dairy cows, 901 (73%) were positive using the ELISA test, and 860 (70%) were positive using the IHA test.

### 3.2. Bayesian Models

Estimated Se and estimated Sp of the ELISA test were similar to their prior values with the medians of 90.5% (95% posterior probability interval (PPI) = 83.2–95.4%) and 70.8% (95% PPI = 60.8–79.8%), respectively (Table 3). On the other hand, the Se of IHA test doubled from its prior value with a medium sensitivity (77.0%, 70.8–84.1 95% PPI), while the Sp of the IHA test had reduced from the prior value (58.2% to 51.1%) (Table 3). The estimate for the HS prevalence was quite high (71.7%, 95% PPI = 62.7–82.6%) and higher than the prior value. The covariance term between the ELISA and IHA tests was lower in both infected- and non-infected animals, with 95% PPI, including 0 regardless of the groups involved. The conditionally independent model, excluding a covariance term between both tests, had a higher DIC value than the conditionally dependent model (29.0 vs. 26.9); therefore, the latter model was considered to be the final model.

After burn-in phase, the model had proper convergence without autocorrelation. Sensitivity analysis demonstrated no appreciable change (changes in median or 95% probability percentiles > 20%) of the posterior estimates of most parameters in the model when a uniform prior non-informative distribution was introduced as a prior value for each parameter. This result confirmed the robustness of the model. Nevertheless, a major change in posterior estimates for the Sp of the IHA test was observed (from 52.3 to 33.1%) when uniform prior non-informative distribution was introduced.

The classical epidemiology calculation demonstrated that the Se and Sp of the ELISA test were 74% and 30%, respectively, both of which are lower than the results from the Bayesian approach.

## 4. Discussion

The present study evaluated the accuracy of a novel indirect ELISA test (a coating antigen derived from *P. multocida* M-1404) and the IHA test (an OIE-recommended assay) for the detection of hemorrhagic septicemia in dairy cows under Thai conditions using a Bayesian approach. The samples for these tests were from herds, which were located in the same area where the management practices were similar. Thus, it was reasonable to consider all dairy cows as a single population, and a one-population model was properly selected for the analysis, as has been assumed in previous studies [4,15].

This study was performed with dairy cow sera from non-vaccinated dairy herds in Chiang Mai province, Thailand. However, the percentages of dairy cows that were positive using the ELISA and IHA tests were found to be high (73 and 70%). This finding suggests that almost 2/3 of the dairy cows in this area possessed antibodies against *P. multocida,* and immunity may have been a result of the recent exposure to the pathogen [5,24].

The posterior estimates of this study demonstrate that the ELISA test had higher sensitivity than the IHA test. This finding agrees with a study in Egypt which suggested that an indirect ELISA test using the outer membrane of *P. multocida* for HS diagnosis revealed higher Se values than the IHA test with percentages of 42%, 92.9%, and 80% for detection in apparently healthy, diseased and emergency slaughtered animals, respectively [25]. The Sp of the ELISA test of the present study was medium (70.8%) and also higher than the IHA test. This result agrees with a previous study in Thailand, which compared the performance of the indirect ELSA and IHA tests on a smaller sample size [4]. Additionally, the study in Egypt suggested that the ELISA test using the capsular antigen of *P. multocida* was a more specific serological test for diagnosis of the HS [25]. Moreover, a study in Iran that involved performing the ELISA on non-vaccinated and non-diseased animals, obtaining a calculation of the Sp from negative animal proportions, revealed that the Sp of an indirect ELISA using *P. multocida* (B:2) outer membrane proteins as an antigen could range from 88.8% to 100% depending on the cut-off values [26]. Other researchers have suggested that antibody titers measuring by an ELISA test were better representatives of protection levels that were achieved after direct challenges. This was because the ELISA test detected only IgG titers. Conversely, IHA and agglutination tests measured the overall IgM and IgG responses; therefore, these tests did not prove to be as reliable as the ELISA test [27].

The estimated prevalence of HS was quite high, and higher than suggested by prior information. The discovery indicated that the true disease prevalence of the dairy cows in this area might be higher than the initial expectation. According to non-vaccinated animals, the high seroprevalence may be due to dairy cows being naturally be exposed to *P. multocida*, or harbored the pathogen in nasopharyngeal regions [25]. However, the authors did not have an individual history of the animals, such as previous diseased experience or previous medication, but most of the herds had experiences of sudden death in cows with undifferentiated respiratory diseases. Most of the farms in this study were supplied by underground water, while a study in Pakistan has demonstrated that supplying underground water on farms increased the risk of outbreaks by 2.9 [28]. It has been discussed that rice paddy crop cultivation may favor the disease outbreaks, whereas the dairy farms in our study were surrounded by rice paddy fields, and buffaloes were pastured in the fields after harvest [29]. Additionally, these farms were smallholders, and cattle movements often occurred to retain their production. This might be spreading the pathogen across the farms. Moreover, Thailand is an endemic area for foot and mouth disease, and a previous study has suggested that the disease increases the risk of HS outbreaks by 2.37 [28]. Although HS is a highly fatal disease, the severity of clinical signs, morbidity, and mortality in dairy cows is lesser than in buffaloes. It has been reported that dairy cows are three times less susceptible than buffaloes [30]. Therefore, the dairy cows in this study could be found a high seroprevalence without clinical signs. This finding is similar to a study in India, which found 25% of seropositive dairy cows with clinical healthy by using a commercial ELISA test [29]. Furthermore, a study in Egypt has demonstrated that the high virulence *P. multocida* isolated from clinically healthy calves, and their seroprevalence reached 42% by using an ELISA test (capsular antigen). Thus, these animals have harbored the virulence pathogen and could be the sources of sporadic outbreaks or act as carriers [25].

A Bayesian latent class analysis is an effective method to solve the problem of an absence of a reference test or a gold standard for estimation of screening test accuracy. Many researchers have conducted analyses to evaluate the Se and Sp values of various diagnostic techniques for many diseases, even in veterinary medicine [9,10,15]. In this study, the Bayesian approach was also performed to evaluate the performance of an in-house indirect ELISA technique (a novel test) compared to the IHA technique (an OIE-recommended assay). Estimated Se and estimated Sp of the ELISA test were close to their prior values. This finding may indicate that the prior distributions were fit for the observed data. Furthermore, the trace plots revealed well mixing of multiple chains of the model, and Gelman-Rubin diagnostic plots showed that the upper limit was close to one. These results indicated the proper convergence of the model. For the model sensitivity analysis, a major change in posterior estimates was only observed in the estimated Sp of the IHA test when uniform prior non-informative distributions were applied. This change suggested that the prior value of this parameter had a strong influence on the posterior estimates of the model. Although the Sp of IHA test in the model was sensitive for the prior selection, the conclusions of the analysis remain unaffected. For example, the use of non-informative prior distributions led to even lower estimates for Sp of the IHA test than estimates of the final model, in which informative prior distributions were assumed. Such results reinforce the main findings of the study, i.e., the ELISA test had higher Se and Sp than the IHA test in the repetition after the disclosure test. Moreover, the Se and Sp of the ELISA test from the classical epidemiology calculation were lower than the estimation of the Bayesian analysis and quite different from the previous validation [4]. This indicates that the Bayesian approach is the reliable method for estimation of the test performance when the reference test (the IHA test) is imperfect.

## 5. Conclusions

In conclusion, this study provides the performance estimation of the indirect ELISA (a novel test) and IHA (an OIE-recommended assay) tests for the detection of immune responses of hemorrhagic septicemia in dairy cows in Chiang Mai province, Thailand. The sensitivity and specificity values obtained from the ELISA test were higher than the IHA test, and the estimated prevalence of HS in the dairy cows in this area was higher than the initial expectation. This finding suggests that an indirect ELISA test using a coating antigen from *P. multocida* M-1404 (B:2) via heat extraction can be a useful tool for the detection of the presence of an antibody against hemorrhagic septicemia in dairy cows.

## Figures and Tables

**Table 1 vetsci-07-00163-t001:** Prior estimates for mode and 95% confidence interval (CI) for sensitivity and specificity values of ELISA test and IHA test, and prevalence of the disease (%).

Diagnostic Tests	Parameters	Mode	95% CI ^a^
ELISA ^b^	Sensitivity	92.1	>87.3
	Specificity	71.9	>63.0
IHA ^c^	Sensitivity	36.0	>26.7
	Specificity	58.2	>50.4
Disease prevalence		35.7	<42.8

^a^ 95% lower or upper credibility interval bound; ^b^ Enzyme-linked immunosorbent assay; ^c^ Indirect hemagglutination assay.

**Table 2 vetsci-07-00163-t002:** Cross-classified test results for hemorrhagic septicemia in 1236 dairy cows from ELISA and IHA tests.

Diagnostic Test	ELISA+ ^a^	ELISA− ^b^	Total
IHA+ ^c^	636	224	860
IHA‒ ^d^	265	111	376
**Total**	901	335	1236

^a^ Number of positive results for the enzyme-linked immunosorbent assay; ^b^ Number of negative results for the enzyme-linked immunosorbent assay; ^c^ Number of positive results for the indirect hemagglutination test; ^d^ Number of negative results for the indirect hemagglutination test.

**Table 3 vetsci-07-00163-t003:** Posterior estimates for median and 95% posterior probability interval (PPI) for sensitivity and specificity of ELISA test and IHA test, and prevalence of the disease (%).

Diagnostic Tests	Parameters	Median	95% PPI ^a^
ELISA ^b^	Sensitivity	90.5	83.2–95.4
	Specificity	70.8	60.8–79.8
IHA ^c^	Sensitivity	77.0	70.8–84.1
	Specificity	51.1	36.8–66.3
Disease prevalence		71.7	62.7–82.6

^a^ 95% Posterior probability interval; ^b^ Enzyme-linked immunosorbent assay; ^c^ Indirect hemagglutination test.

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
