# Peer review of "Validation of a Novel ELISA for the Diagnosis of Hemorrhagic Septicemia in Dairy Cattle from Thailand Using a Bayesian Approach"

_vetsci, 2020, doi:10.3390/vetsci7040163_

Round 1

Reviewer 1 Report

The revised version of the paper satisfactorily addresses the issues raised against the previous version. My review has found a few, very minor, issues that should be handled, but none that question the validity, results or conclusions of the paper.

Minor issues:

1. Lines 18-22: In the abstract, something has gone wrong with the summary of the numeric results, with unclosed parentheses and mixed-up words. Please restore this along the lines of the previous version.

2. Lines 121-122: It is suggested (but not required) to help the reader recognize that the 97.5% originates from a 95% interval. This could for instance be in the form of a parenthesis like this: “... a 97.5% lower limit for prior distributions (i.e. above the lower limit of the 95% interval for the prior).”

3. Line 130: Should not “test” be “assay”? IHA = Indirect hemagglutination assay.

4. Line 144: 860 out of 1,236 is reported as “67%”, but should be 70% (860/1,236 = 0.696). To help the reader recognize the numbers, the authors may also want to round “(72.9 and 69.6%)” to “(73 and 70%)” in line 183.

The English of the paper is good. A few grammatical and typographical have been noted, though, all of which can easily be corrected.

Lines 15 and 107-108: For some reason, the English language does not use the definite article before “both”, so “the both tests” should be “both tests” (but grammar can be a strange thing, “the two tests” is correct).

Line 119: “showing in Table 1” -> “shown in Table 1”

Line 152: There is a double parenthesis after the abbreviation, “(PPI))”

Line 155: “Estimate” -> “The estimate”

Line 169: The role of the word “only” is unclear. It could be present in a sentence of the form “was only observed when” (followed by the condition of when it was observed). In this line, it seems like “only” is not needed.

Line 172: Suggestion: “which lower” -> “both of which are lower”

Line 219: “less susceptible to buffaloes” -> “less susceptible than buffaloes” (both cows and buffaloes are susceptible to HS, but cows are less susceptible less than buffaloes)

Line 249: “In conclusions” -> “In conclusion” (single, not plural, even if more than one conclusion follows – another grammatical strangeness. While the paragraph headline “Conclusions” is plural as expected.)

Author Response

Authors Responses to the Reviewer(s)' Comments to Author:

Manuscript number: vetsci-982892

Manuscript title: Validation of a novel ELISA for the diagnosis of haemorrhagic septicaemia in dairy cattle from Thailand using a Bayesian approach.

Article type: Research Paper

Journal: Veterinary Sciences

Authors: Tawatchai Singhla, Pallop Tankaew, Nattawooti Sthitmatee

Response to reviewer 1 comments (Round 1):

  1. Lines 18-22: In the abstract, something has gone wrong with the summary of the numeric results, with unclosed parentheses and mixed-up words. Please restore this along the lines of the previous version.

Response: We thank the reviewer for the valuable suggestions. We apologize for these errors. The sentences have been corrected on line 17-21 (with a yellowing highlighted text).  

            “Estimated sensitivity and estimated specificity of the ELISA test were 90.5% [95% posterior probability interval (PPI) = 83.2-95.4%] and 70.8% (95% PPI = 60.8-79.8%), respectively. Additionally, estimates for the Se and Sp values of the IHA test were 77.0% (95% PPI =70.8-84.1%) and 51.1% (PPI= 36.8-66.3%), respectively. The estimated prevalence of the disease was 71.7% (95% PPI= 62.7-82.6%).”

  1. Lines 121-122: It is suggested (but not required) to help the reader recognize that the 97.5% originates from a 95% interval. This could for instance be in the form of a parenthesis like this: “... a 97.5% lower limit for prior distributions (i.e. above the lower limit of the 95% interval for the prior).”

Response: Thank you very much for the kind suggestion. We agree with the reviewer. This point has been added on line 122 (with a yellowing highlighted text).  

            “a 97.5% lower limit for prior distributions (i.e. above the lower limit of the 95% interval for the prior).”

  1. Line 130: Should not “test” be “assay”? IHA = Indirect hemagglutination assay.

Response: We thank the reviewer for the kind suggestions. We agree with the reviewer. The word has been edited on line 130 (with a yellowing highlighted text).  

            “c Indirect hemagglutination assay.”

  1. Line 144: 860 out of 1,236 is reported as “67%”, but should be 70% (860/1,236 = 0.696). To help the reader recognize the numbers, the authors may also want to round “(72.9 and 69.6%)” to “(73 and 70%)” in line 183.

Response: Thank you very much for the kind suggestion. We agree with the reviewer. The words have been edited on line 145 and 184 (with a yellowing highlighted text).  

            “Of 1,236 dairy cows, 901 (73%) were positive to the ELISA test, and 860 (70%) were positive to the IHA test.”

            “IHA tests were found to be high (73 and 70%).”

The English of the paper is good. A few grammatical and typographical have been noted, though, all of which can easily be corrected.

Lines 15 and 107-108: For some reason, the English language does not use the definite article before “both”, so “the both tests” should be “both tests” (but grammar can be a strange thing, “the two tests” is correct).

Response: We thank the reviewer for the kind suggestions. We agree with the reviewer. The words have been edited on line 15 and 107-108 (with a yellowing highlighted text).  

            “from 44 farms were tested with the two tests”

            “The reaction mechanism of the two tests was based on the specific antibody detection”

Line 119: “showing in Table 1” -> “shown in Table 1”

Response: Thank you very much for the kind suggestion. We agree with the reviewer. The word has been edited on line 119 (with a yellowing highlighted text).  

            “The prior values of all parameters (Se, Sp and prevalence), shown in Table 1,”

Line 152: There is a double parenthesis after the abbreviation, “(PPI))”

Response: We thank the reviewer for the kind suggestions. We agree with the reviewer. The error has been corrected on line 153 (with a yellowing highlighted text).  

            “[95% posterior probability interval (PPI) =83.2–95.4%]”

Line 155: “Estimate” -> “The estimate”

Response: Thank you very much for the kind suggestion. We agree with the reviewer. The word has been edited on line 156 (with a yellowing highlighted text).

                “The estimate for the HS prevalence was quite high (71.7%, 95% PPI= 62.7–82.6%)”   

Line 169: The role of the word “only” is unclear. It could be present in a sentence of the form “was only observed when” (followed by the condition of when it was observed). In this line, it seems like “only” is not needed.

Response: We thank the reviewer for the kind suggestions. We agree with the reviewer. The word has been removed on line 170 (with a yellowing highlighted text).  

            “IHA test was observed (from 52.3 to 33.1%)”

Line 172: Suggestion: “which lower” -> “both of which are lower”

Response: We thank the reviewer for the kind suggestions. We agree with the reviewer. The sentence has been edited on line 173 (with a yellowing highlighted text).

                “both of which are lower than the results”  

Line 219: “less susceptible to buffaloes” -> “less susceptible than buffaloes” (both cows and buffaloes are susceptible to HS, but cows are less susceptible less than buffaloes)

Response: Thank you very much for the kind suggestion. We agree with the reviewer. The sentence has been edited on line 220 (with a yellowing highlighted text).

                “dairy cows are three times less susceptible than buffaloes”

Line 249: “In conclusions” -> “In conclusion” (single, not plural, even if more than one conclusion follows – another grammatical strangeness. While the paragraph headline “Conclusions” is plural as expected.)

Response: We thank the reviewer for the kind suggestions. We agree with the reviewer. The sentence has been edited on line 250 (with a yellowing highlighted text).

                “In conclusion, this study provides the performance estimation of the indirect ELISA (a novel test) and IHA (an OIE-recommended assay) tests”  

Reviewer 2 Report

The authors provided answers to all my comments and corrected the manuscript accordingly. The manuscript has some limitations but its conclusions are supported by the results and this information should be presented to the scientific community. Therefore I recommend the publication of the manuscript. 

Author Response

We thank the reviewer for the valuable comments and suggestions. Those made the manuscript better.

This manuscript is a resubmission of an earlier submission. The following is a list of the peer review reports and author responses from that submission.

Round 1

Reviewer 1 Report

The paper by Dr. Singla, Dr. Tankaew, and Dr. Sthitmatee sets out to determine sensitivity and specificity of a novel ELISA test along with an established IHA test for the cattle disease haemorrhagic septicaemia as well as the prevalence of the disease. The sample material is from 1,236 samples from 44 farms in Thailand. As the existing (IHA) reference test is know to be imperfect, a Bayesian approach is used, incorporating both prior knowledge from earlier studies as well as the data from the present study. The results show ELISA to have sensitivity of 90.5% and specificity 70.8% (which is similar to the prior), while the IHA was found to have sensitivity 77.0% (markedly higher than the prior) and specificity 51.1% (somewhat lower than the prior). Disease prevalence was estimated as 71.7%, twice as high as the prior of 35.7%. The authors conclude that the ELISA test can be a useful tool for detection of haemorrhagic septicaemia, which appears to have high prevalence in the area.

The paper presents interesting results that validates the relevance of the novel ELISA test. The Bayesian approach seems like a good choice in a situation where no solid reference test is available.

Major issues:

1. References to [4] (OIE manual) seems in several (all?) cases to refer to the earlier study [3] (Tankaew et al, 2018). Please check that reference numbers correspond to numbers in the reference list.

2. Comparing tables 1 and 3, IHA sensitivity drops to half its value from prior to posterior, while disease prevalence doubles. The discussion of these changes is very sparse and should be augmented.

Minor issues:

3. Line 101: As there are only four different cases when two tests are compared, it is suggested to replace “e.g. +/+, +/-, etc.” with a full list: “+/+, +/-, -/+, or -/-”.

4. Lines 105-106: According to the text, the prior data were based on expert opinion and earlier findings fond in ref. [4]. Assuming that the meaning is ref. [3], the numbers are no longer expert opinions, but simply the earlier finding (posterior values from the earlier study), even if the earlier study included expert opinion as part of the input. Agree?

5. Lines 107-108: “the lowest modal value was defined as a 95% lower limit for prior distributions”. Is this to be understood as the limit above which 95% of the distribution will be? Or is it the lower limit of a symmetrical 95% prior interval? The latter would be from 2.5% to 97.5%, thus having 97.5% of the distribution above the lower limit (and the same percentage below the upper limit).

6. Table 1. Comparing the rightmost column in Table 1 with ref. [3], there are minor differences in the decimals (87.5 or 87.3? 43.0 or 42.8?). These decimals most likely carry no real information, but please check that the stated numbers correspond to the reference.

7. Table 1: The last column, “95% CI” is described as “95% lower and upper credibility interval bound”. However, the column does not contain an interval, only one bound (lower for Se and Sp, upper for disease prevalence) in each case. Please make the description reflect this, e.g. by writing “upper or lower” instead of “lower and upper”.

8. Table 2: It is suggested to augment the table with sums, both horizontally and vertically. E.g.:

IHA+ 636 224 860

(636 + 224 = 860).

9. Lines 124-132: The paragraph text is to a large extent repetition of the data presented in Table 2. It is suggested to keep the main numbers in Table 2, and then give an overview (without full repetition) in the text. Suggestion for inspiration:

“The test results from tests are summarized in Table 2. Of the 1,236 dairy cows, the ELISA test was positive in 73% and negative in 27% of the cases (901/1,236 and 335/1,236), while the IHA test was positive in 70% and negative in 30% of the cases (860/1,236 and 376/1,236). The two tests had 60% overall agreement.”

(This is just a suggestion, the text could be formulated in many ways. The point is to summarize rather than repeat. In the example, the percentages have been rounded, as the decimal carries very little information above stochastic noise).

10. Line 137: Se and Sp of the ELISA test is described as “slightly lower” than their prior values. Technically this is true (Se changed from 92.1 to 90.5; Sp from 71.9 to 70.8), but the difference is considerably smaller than the with of the probability interval, and as such it is suggested to write “similar to” or “close to” (wording used in line 197) rather than “slightly lower”.

11. Line 140: Characteristics (Se and Sp) of the IHA is described as “similar to” the prior values. This should be rephrased. Se has doubled from 36.0 to 77.0, Sp has been somewhat reduced from 58.2 to 51.1.

12. Table 3: There seems to be an error in the ELISA 95% PPI. According to Table 3, the lower PPI boundary for specificity is equal to the median (70.8). According to line 139, the lower PPI boundary should be 60.8 instead of 70.8. Please check.

14. Lines 197-198: Noting that Se and Sp for the ELISA were close to their prior values, this is described with the sentence: “This finding may indicate that the model had a lack of identifiability and the prior distributions were fit for the observed data.” Is this to be understood as two opposing interpretations (it could be due to a lack of identifiability, or it could be because the prior already fit the data well) or as one compound interpretation (there is lack of identifiability due to the prior fitting the data well)? In any case, the term “lack of identifiability” seems to indicate that the data cannot really be trusted. If this is the intended meaning, please elaborate. If it is not (the present study contains 1,236 tests, which is about three times as much as the 415 samples in ref. [3]), please consider rephrasing to avoid misunderstanding.

Author Response

Authors Responses to the Reviewer(s)' Comments to Author:

Manuscript number: vetsci-922394

Manuscript title: Estimation of sensitivity and specificity of a novel ELISA and an indirect hemagglutination for detection of specific antibody against haemorrhagic septicaemia in dairy cattle in Thailand using a Bayesian approach

Article type: Research Paper

Journal: Veterinary Sciences

Authors: Tawatchai Singhla, Pallop Tankaew, Nattawooti Sthitmatee

Response to reviewer 1 comments:

Major issues:

Comment 1: References to [4] (OIE manual) seems in several (all?) cases to refer to the earlier study [3] (Tankaew et al, 2018). Please check that reference numbers correspond to numbers in the reference list.

Response: Thank you very much for mentioning this error point. The references to [4] (OIE manual) and [3] (Tankaew et al., 2018) have been checked and the error was automatic generation of reference list number (number 1 of the reference list was double). The correct reference list numbers have applied which start from line 259.

Comment 2: Comparing tables 1 and 3, IHA sensitivity drops to half its value from prior to posterior, while disease prevalence doubles. The discussion of these changes is very sparse and should be augmented.

Response: We thank the reviewer for the comments. However, IHA sensitivity increased to half from prior to posterior estimates (36% to 77%). Therefore, the prevalence of the disease was increased as well. The data has shown in Table 1 (line 127) and Table 3 (line 158).

Minor issues:

Comment 3: Line 101: As there are only four different cases when two tests are compared, it is suggested to replace “e.g. +/+, +/-, etc.” with a full list: “+/+, +/-, -/+, or -/-”.

Response: We thank the reviewer for the comments. We agree with the reviewer. Thus, appropriate sentence of the different combinations of the test result has been replaced on line 114-115. (with a yellowing highlighted text).

            “The Bayesian latent class model defined that for the k populations, the counts (Yk) of the different combinations of test results such as +/+, +/−, -/+, or -/- for the two tests followed a multinomial distribution: Yk | Pqrk ~ multinomial (nk, {Pqrk}), where qr was the multinomial cell probability for the two-test outcome combination, and Pqrk was a vector of the probabilities of observing the individual combinations of the test results.”

Comment 4: Lines 105-106: According to the text, the prior data were based on expert opinion and earlier findings fond in ref. [4]. Assuming that the meaning is ref. [3], the numbers are no longer expert opinions, but simply the earlier finding (posterior values from the earlier study), even if the earlier study included expert opinion as part of the input. Agree?

Response: Thank you for the kind comments. We agree with the reviewer. The prior data was based on only the previous study in ref. [4] because the expert opinion was included in the earlier study. Thus, the words “expert opinion” has been removed from the sentence which shows on line 118-120. (with a yellowing highlighted text).

            “The prior values of all parameters (Se, Sp and prevalence), showing in Table 1, were introduced to the model using beta distribution based on the findings of a previous report [4].” 

Comment 5: Lines 107-108: “the lowest modal value was defined as a 95% lower limit for prior distributions”. Is this to be understood as the limit above which 95% of the distribution will be? Or is it the lower limit of a symmetrical 95% prior interval? The latter would be from 2.5% to 97.5%, thus having 97.5% of the distribution above the lower limit (and the same percentage below the upper limit).

Response: We thank the reviewer for the comments. The sentence “a 95% lower limit for prior distributions” means the lower limit of a symmetrical 95% prior interval. Thus, we have changed “a 95% lower limit” to “a 97.5% lower limit” on line 120-122. (with a yellowing highlighted text).

            “The means of the central values provided in this study were chosen as being the most likely values, while the lowest modal value was defined as a 97.5% lower limit for prior.”

 Comment 6:  Table 1. Comparing the rightmost column in Table 1 with ref. [3], there are minor differences in the decimals (87.5 or 87.3? 43.0 or 42.8?). These decimals most likely carry no real information, but please check that the stated numbers correspond to the reference.  

Response: Thank you for mentioning this typo. We have changed 87.5 to 87.3 and 43.0 to 42.8 on line 127. (with a yellowing highlighted text).

Comment 7:  Table 1: The last column, “95% CI” is described as “95% lower and upper credibility interval bound”. However, the column does not contain an interval, only one bound (lower for Se and Sp, upper for disease prevalence) in each case. Please make the description reflect this, e.g. by writing “upper or lower” instead of “lower and upper”.

Response: We thank the reviewer for your guidance. We agree with the reviewer. Thus, the word “lower and upper” has been replaced with “lower or upper” as shown on line 129. (with a yellowing highlighted text).

Comment 8: Table 2: It is suggested to augment the table with sums, both horizontally and vertically. E.g.: IHA+ 636 224 860 (636 + 224 = 860).  

Response: We thank the reviewer for the guidance. We agree with the reviewer. The total numbers of each column and row have been added on line 142. (with a yellowing highlighted text).  

Comment 9: Lines 124-132: The paragraph text is to a large extent repetition of the data presented in Table 2. It is suggested to keep the main numbers in Table 2, and then give an overview (without full repetition) in the text. Suggestion for inspiration:

“The test results from tests are summarized in Table 2. Of the 1,236 dairy cows, the ELISA test was positive in 73% and negative in 27% of the cases (901/1,236 and 335/1,236), while the IHA test was positive in 70% and negative in 30% of the cases (860/1,236 and 376/1,236). The two tests had 60% overall agreement.”

(This is just a suggestion, the text could be formulated in many ways. The point is to summarize rather than repeat. In the example, the percentages have been rounded, as the decimal carries very little information above stochastic noise).

Response: Thank you very much for the valuable comments. We agree with these comments and suggestions. We have changed those paragraph texts according to your suggestions on line 140-141. (with a yellowing highlighted text).  

             “The diagnostic test results are shown in Table 2. Of 1,236 dairy cows, 901 (73%) were positive to the ELISA test, and 860 (67%) were positive to the IHA test.”

Comment 10: Line 137: Se and Sp of the ELISA test is described as “slightly lower” than their prior values. Technically this is true (Se changed from 92.1 to 90.5; Sp from 71.9 to 70.8), but the difference is considerably smaller than the with of the probability interval, and as such it is suggested to write “similar to” or “close to” (wording used in line 197) rather than “slightly lower”.

Response: We thank the reviewer for the comments. We agree with these comments. The word “slightly low” has been replaced with “similar to” on line 148. (with a yellowing highlighted text).  

            “Estimated Se and estimated Sp of the ELISA test were similar to their prior values.”

Comment 11: Line 140: Characteristics (Se and Sp) of the IHA is described as “similar to” the prior values. This should be rephrased. Se has doubled from 36.0 to 77.0, Sp has been somewhat reduced from 58.2 to 51.1.

Response: Thank you very much for the suggestions. We agree with your suggestions. The sentence has been changed according to your suggestions on line 150-152. (with a yellowing highlighted text).  

            “On the other hand, the Se of IHA test has doubled from its prior value with a medium sensitivity (77.0%, 70.8–84.1 95% PPI), while the Sp of the IHA test has reduced from the prior value (58.2% to51.1%).”

Comment 12: Table 3: There seems to be an error in the ELISA 95% PPI. According to Table 3, the lower PPI boundary for specificity is equal to the median (70.8). According to line 139, the lower PPI boundary should be 60.8 instead of 70.8. Please check.

Response: We thank the reviewer for mentioning this error point. The error has been corrected on line 158. (with a yellowing highlighted text).  

Comment 13: Lines 197-198: Noting that Se and Sp for the ELISA were close to their prior values, this is described with the sentence: “This finding may indicate that the model had a lack of identifiability and the prior distributions were fit for the observed data.” Is this to be understood as two opposing interpretations (it could be due to a lack of identifiability, or it could be because the prior already fit the data well) or as one compound interpretation (there is lack of identifiability due to the prior fitting the data well)? In any case, the term “lack of identifiability” seems to indicate that the data cannot really be trusted. If this is the intended meaning, please elaborate. If it is not (the present study contains 1,236 tests, which is about three times as much as the 415 samples in ref. [3]), please consider rephrasing to avoid misunderstanding.

Response: We thank the reviewer for the guidance. We agree with these comments. The sentence has been rephrased on line 227-228.

            “This finding may indicate that the prior distributions were fit for the observed data.”

Reviewer 2 Report

The manuscript Estimation of sensitivity and specificity of a novel ELISA and an indirect hemagglutination for detection of specific antibody against haemorrhagic septicaemia in dairy cattle in Thailand using a Bayesian approach from Singhla and others presents unexpected results of 70% antibodies prevalence through a novel indirect ELISA test. The main problem in my opinion was the use of IHA test to validate the ELISA using a Bayesian approach. The reported Se for IHA test is quite low, 36%, I while de Sp is 58.2. The author said that the prior value of Sp and Se had a strong influence on the posterior estimates of the model. To valid this novel ELISA, they should have use know cases (past) and control animals. I have some doubts regarding the validity of the data. This results maybe be false positive due cross-reaction. Haemorrhagic septicaemia is a disease with a high mortality rate. How it would be possible a prevalence of 72%? This must be discussed. There is limited information about the condition of the sampled animals. An antibody test gives the information that the animal had contact with the pathogen. In several cases, it say that the animal already had the infection. Considering the mortality rate of HS, what this positive results say. What is the information from the diagnostic test. There were animals that have the disease and recovered after? They were vaccinated. Please, address this issue in the revised manuscript. Since the OIE recommends the IHA as diagnostic test, I would like to see the Se and Sp results for the ELISA compared with IHA as gold standard. Theses results would improve your discussion. Additional comments: There are recent literature missing, the manuscript has limited literature. https://doi.org/10.1080/20008686.2019.1604064 https://doi.org/10.1016/j.sjbs.2015.06.011 The tittle is too long, I recommend the authors to reduce it. Replace Estimation of sensitivity and specificity for validation Remove specific antibody against Suggestion: Validation of a novel ELISA for the diagnosis of haemorrhagic septicaemia in dairy cattle from Thailand using a Bayesian approach. In the abstract you don´t specified the gold standard used for the Se and Sp analysis. Thus, you must describe the method used to analyze the data (Bayesian) PPI? Do you mean confidence internal? How do you estimate the Se and Sp for IHA? L25: Notably, the cows in this area indicated a high percentage of exposure to a variety of pathogens. Variety of pathogens? You test only to one. L35 to 37. I did not understood what you mean. Please rephrase. You are talking about economical losses? L65-78: What about the positive control and cut-off calculation. Please add such important information. L90: The disease was suggested if titers from 1:160 up to 1:1280 or higher were detected The sample was considered positive at IHA test if…. L124-126: be more clear, you don´t need to inform the negative. From the 1,236 dairy cowas xxx (%) were positive to the ELISA and XXX (%) were positive to the IHA test. L167: This finding suggests that more than half of Too vague, almost 2/3 would be also correct. Please rephrase. L182: B:2 Sometimes (B:2). Be consistent.

Author Response

Authors Responses to the Reviewer(s)' Comments to Author:

Manuscript number: vetsci-922394

Manuscript title: Estimation of sensitivity and specificity of a novel ELISA and an indirect hemagglutination for detection of specific antibody against haemorrhagic septicaemia in dairy cattle in Thailand using a Bayesian approach

Article type: Research Paper

Journal: Veterinary Sciences

Authors: Tawatchai Singhla, Pallop Tankaew, Nattawooti Sthitmatee

Response to reviewer 2 comments:

 Comment 1: The main problem in my opinion was the use of IHA test to validate the ELISA using a Bayesian approach. The reported Se for IHA test is quite low, 36%, I while de Sp is 58.2. The author said that the prior value of Sp and Se had a strong influence on the posterior estimates of the model.

Response: We thank the reviewer for the comments. This point is really importance. For the model sensitive analysis, only prior value of Sp of the IHA test had a strong influence on the posterior estimates of the model, whereas the estimated Se was similar to its prior value. The estimated Sp of the IHA test was changed from 51.1 % to 33.1% after the Sp prior value was replaced by a non-informative uniform 0-1 distribution (non-informative prior value). Furthermore, we also have analyzed our observed data with prior values from previous study which defined the Se and Sp of the IHA from expert opinions as 90% and 80%, respectively. After analysis, the estimated Se and Sp were 76.9% and 49.4%, respectively, and estimated Sp of IHA was only observed influence of the prior value as well. Therefore, this result was still similar to the result in the manuscript (Se = 77.0% and Sp = 51.1%), and prior selection in this study was quite reasonable. These have been added to the discussion on line 230-238. (with a yellowing highlighted text).  

            “For the model sensitivity analysis, a major change in posterior estimates was only observed in the estimated Sp of the IHA test when uniform prior non-informative distributions were applied. This change suggested that the prior value of this parameter had a strong influence on the posterior estimates of the model. Although the Sp of IHA test in the model was sensitive for the prior selection, the conclusions of the analysis remain unaffected. For example, use of non-informative prior distributions led to even lower estimates for Sp of the IHA test than estimates of the final model which informative prior distributions were assumed. Such results reinforce the main findings of the study, i.e., the ELISA test had higher Se and Sp than the IHA test in the first repetition after the disclosure test.”

Comment 2: To valid this novel ELISA, they should have use know cases (past) and control animals.

Response: Thank you very much for the kind suggestion. The ELISA test was validated in the previous study ref. [4] that they used healthy dairy cow sera, immunizing with a formaldehyde-fixed P. multocida, as positive controls and colostrum-deprived neonatal calf sera as a negative serum control.  These have been explained online 43-47. (with a yellowing highlighted text).  

            “Recently, Thai researchers have developed an indirect ELISA test to detect specific antibodies against P. multocida in dairy cows using a coating antigen from P. multocida M-1404 (B:2) heat extraction [4]. The researchers have validated the ELISA test in positive and negative control sera and suggested that this method has displayed reliable degrees of performance with Se and Sp values of 92.1 and 71.9, respectively.”

Comment 3: I have some doubts regarding the validity of the data. This results maybe be false positive due cross-reaction. Haemorrhagic septicaemia is a disease with a high mortality rate. How it would be possible a prevalence of 72%? This must be discussed.

Response: Thank you very much for valuable comments. Haemorrhagic septicaemia affects animal health with high morbidity and mortality, especially buffaloes. However, clinical signs and mortality in dairy cows are lower than buffaloes. The high prevalence has been discussed on line 198-220. (with a yellowing highlighted text).  

            “Estimated prevalence of the HS was quite high and higher than the prior information. The discovery indicated that the true disease prevalence of the dairy cows in this area may be higher than initial expectation. According to non-vaccinated animals, the high seroprevalence may be due to that the dairy cows might naturally be exposed to P. multocida or harbored the pathogen in nasopharyngeal regions [25]. However, the authors did not have individual history of animal such as previous diseased experience or previous medication, but most of herds had experiences of sudden death cows with undifferentiated respiratory diseases. Most of farms in this study are supplied by underground water, while the study in Pakistan has demonstrated that supplying underground water on farms increased risk of outbreaks by 2.9 [28]. It has been discussed that rice paddy crop cultivation may favor the disease outbreaks, whereas the dairy farms in our study were surrounded with rice paddy fields, and buffaloes were pastured in the fields after harvest [29]. Additionally, these farms were smallholders, and cattle movements were often occurred to remain their productions. This might be spreading of the pathogen across the farms. Moreover, Thailand is endemic area of foot and mouth disease, and a previous study has suggested that the disease increase the risk of HS outbreaks by 2.37 [28]. Although the HS is highly fatal disease, severity of clinical signs, morbidity and mortality in dairy cows are lesser than buffaloes. It has been reported that dairy cows are three times less susceptible to buffaloes [30]. Therefore, the dairy cows in this study could be found a high seroprevalence without clinical signs. This finding is similar to the study in India which found 25% of seropositive dairy cows with clinical healthy by using a commercial ELISA test [29]. Furthermore, the study in Egypt has demonstrated that the high virulence P. multocida were isolated from clinical healthy calves, and their seroprovalence reached to 42% by using ELISA test (capsular antigen). Thus, these animals have harbored the virulence pathogen and could be the sources of the sporadic outbreak or carriers [25].”  

Comment 4: There is limited information about the condition of the sampled animals. An antibody test gives the information that the animal had contact with the pathogen. In several cases, it say that the animal already had the infection. Considering the mortality rate of HS, what this positive results say. What is the information from the diagnostic test. There were animals that have the disease and recovered after? They were vaccinated. Please, address this issue in the revised manuscript.

Response: Thank you very much for valuable comments. We agree with the reviewer. The positive results, information from the diagnostic test, diseased experience of animals and vaccination of animals have discussed on line 198-220. (with a yellowing highlighted text).  

            “Estimated prevalence of the HS was quite high and higher than the prior information. The discovery indicated that the true disease prevalence of the dairy cows in this area may be higher than initial expectation. According to non-vaccinated animals, the high seroprevalence may be due to that the dairy cows might naturally be exposed to P. multocida or harbored the pathogen in nasopharyngeal regions [25]. However, the authors did not have individual history of animal such as previous diseased experience or previous medication, but most of herds had experiences of sudden death cows with undifferentiated respiratory diseases. Most of farms in this study are supplied by underground water, while the study in Pakistan has demonstrated that supplying underground water on farms increased risk of outbreaks by 2.9 [28]. It has been discussed that rice paddy crop cultivation may favor the disease outbreaks, whereas the dairy farms in our study were surrounded with rice paddy fields, and buffaloes were pastured in the fields after harvest [29]. Additionally, these farms were smallholders, and cattle movements were often occurred to remain their productions. This might be spreading of the pathogen across the farms. Moreover, Thailand is endemic area of foot and mouth disease, and a previous study has suggested that the disease increase the risk of HS outbreaks by 2.37 [28]. Although the HS is highly fatal disease, severity of clinical signs, morbidity and mortality in dairy cows are lesser than buffaloes. It has been reported that dairy cows are three times less susceptible to buffaloes [30]. Therefore, the dairy cows in this study could be found a high seroprevalence without clinical signs. This finding is similar to the study in India which found 25% of seropositive dairy cows with clinical healthy by using a commercial ELISA test [29]. Furthermore, the study in Egypt has demonstrated that the high virulence P. multocida were isolated from clinical healthy calves, and their seroprovalence reached to 42% by using ELISA test (capsular antigen). Thus, these animals have harbored the virulence pathogen and could be the sources of the sporadic outbreak or carriers [25].”

However, information of animals such as vaccination has also been stated in materials and methods on line 66-70. (with a yellowing highlighted text).  

            “The dairy cow sera (n = 1236) from 44 farms which located in Sanpatong, Mae-Wang and Doi-Loh districts, Chiang Mai province were obtained from the Chiang Mai Provincial Livestock Office, the Department of Livestock Development (DLD), Thailand. Serum samples were collected from non-vaccinated dairy cows (age ≥ 1 year) by the DLD annual disease investigation program regardless of health and production status.”

Comment 5: Since the OIE recommends the IHA as diagnostic test, I would like to see the Se and Sp results for the ELISA compared with IHA as gold standard. These results would improve your discussion.

Response: We thank the reviewer for the comments and providing us of the recent studies. This point is importance. We have analyzed according to the reviewer suggestions and found that the sensitivity of the ELISA test was 74%, and the specificity of the ELISA was 30% when compared with the IHA test as the gold standard.

Comment 6: The tittle is too long, I recommend the authors to reduce it. Replace Estimation of sensitivity and specificity for validation Remove specific antibody against Suggestion: Validation of a novel ELISA for the diagnosis of haemorrhagic septicaemia in dairy cattle from Thailand using a Bayesian approach.

Response: Thank you very much for your kind suggestion. We agree with the reviewer that the title is too long. This helps us a lot. The title has been changed in “Validation of a novel ELISA for the diagnosis of haemorrhagic septicaemia in dairy cattle from Thailand using a Bayesian approach.”

Comment 7: In the abstract you don´t specified the gold standard used for the Se and Sp analysis. Thus, you must describe the method used to analyze the data (Bayesian) PPI? Do you mean confidence internal?

Response: We thank the reviewer for the comments. We really realized this point. According to the limitation of the journal, the abstract should be a total of about 200 words maximum. Therefore, we cannot add information more that. The limitation of this study was the absence of the gold standard because no any method has 100% of Se and Sp for the disease testing. Therefore, we used the Bayesian approach to solve this problem. The results of the Bayesian approach are expressed in posterior medians (sensitivity, specificity and prevalence) and their 95% posterior probability interval (PPI). The latter term is used in a Bayesian statistic instead a confident interval in frequentist statistic.  However, the Bayesian method has been explained in the introduction and the materials and methods parts on line 49-58 and 106-124. (with a yellowing highlighted text).  

            “Determination of the sensitivity and specificity of a screening test is usually done by comparing to a reference test (a gold standard) or applying to true infected animals and true healthy animals. The sensitivity and specificity can be estimated directly if a gold standard is available [6] However, the perfect reference test and large population of the true infected animals are unavailable. These are the major problems for estimation of diagnostic test performance in this study. Moreover, serious bias may be introduced in the evaluation of the accuracies of the novel test when classification errors in the reference test are overlooked [7, 8]. Thus, a Bayesian latent class analysis was performed to address these problems because this method has inferred posterior estimates from combination of prior information and observed data to solve uncertainty of unknown parameters such as test performance or disease prevalence.”

            “A Bayesian latent class modeling was conducted to estimate Se and Sp values of the indirect-ELISA and IHA tests as described in the previous reports [9, 14]. The reaction mechanism of the both tests was based on the specific antibody detection; therefore, the results of the tests were defined as conditional dependence upon each other [14]. The sera were assumed that they were from the same population because these samples were obtained from farms which located in the same area as suggested in the previous studies [10, 15]. Therefore, a Bayesian model for two conditionally dependent tests and one population was adjusted to the observed data for estimation of the Se and Sp values of the two tests and the true disease prevalence.

            The Bayesian latent class model defined that for the k populations, the counts (Yk) of the different combinations of test results such as +/+, +/−, -/+, or -/- for the two tests followed a multinomial distribution: Yk | Pqrk ~ multinomial (nk, {Pqrk}), where qr was the multinomial cell probability for the two-test outcome combination, and Pqrk was a vector of the probabilities of observing the individual combinations of the test results. The prior values of all parameters (Se, Sp and prevalence), showing in Table 1, were introduced to the model using beta distribution based on the findings of a previous report [4]. The means of the central values provided in this study were chosen as being the most likely values, while the lowest modal value was defined as a 97.5% lower limit for prior distributions. After analysis, the median of the posterior distributions was used for point estimates. The standard deviation of a marginal posterior distribution referred to 95% posterior probability interval (PPI) [4, 11].”

Comment 8: How do you estimate the Se and Sp for IHA?

Response: We thank the reviewer for the comments. This is a good point. In general, the Bayesian statistic simulates posterior estimates from prior values (data from previous studies or expert opinions) and observed data (test results) via Markov chain Monte Carlo simulation. Therefore, we can estimate the sensitivity, specificity and prevalence of a disease from this simulation. These have been added in the materials and methods on line 106-124. (with a yellowing highlighted text).  

            “A Bayesian latent class modeling was conducted to estimate Se and Sp values of the indirect-ELISA and IHA tests as described in the previous reports [9, 14]. The reaction mechanism of the both tests was based on the specific antibody detection; therefore, the results of the tests were defined as conditional dependence upon each other [14]. The sera were assumed that they were from the same population because these samples were obtained from farms which located in the same area as suggested in the previous studies [10, 15]. Therefore, a Bayesian model for two conditionally dependent tests and one population was adjusted to the observed data for estimation of the Se and Sp values of the two tests and the true disease prevalence.

            The Bayesian latent class model defined that for the k populations, the counts (Yk) of the different combinations of test results such as +/+, +/−, -/+, or -/- for the two tests followed a multinomial distribution: Yk | Pqrk ~ multinomial (nk, {Pqrk}), where qr was the multinomial cell probability for the two-test outcome combination, and Pqrk was a vector of the probabilities of observing the individual combinations of the test results. The prior values of all parameters (Se, Sp and prevalence), showing in Table 1, were introduced to the model using beta distribution based on the findings of a previous report [4]. The means of the central values provided in this study were chosen as being the most likely values, while the lowest modal value was defined as a 97.5% lower limit for prior distributions. After analysis, the median of the posterior distributions was used for point estimates. The standard deviation of a marginal posterior distribution referred to 95% posterior probability interval (PPI) [4, 11].”

Comment 9: L25: Notably, the cows in this area indicated a high percentage of exposure to a variety of pathogens. Variety of pathogens? You test only to one.

Response: Thank you the reviewer for your kind suggestion. We agree with the reviewer. The words “variety of the pathogen” have been replaced by “Pasteurella multocida” on line 23-24. (with a yellowing highlighted text).  

            “Notably, the cows in this area indicated a high percentage of exposure to Pasteurella multocida.

Comment 10: L35 to 37. I did not understood what you mean. Please rephrase. You are talking about economical losses?

Response: We thank the reviewer for the comments. That point is about economic losses. The sentence means that the economic losses of the disease outbreak were from direct losses (mortality losses) and indirect losses (treatment costs, growth reduction and low milk yields). The sentence has been rephrased on line 33-35. (with a yellowing highlighted text).  

            “It has been reported that mortality losses (direct losses) account for 92.7% of total losses, while the rest (7.3%) were due to treatment costs, growth reduction and low milk yields [3].”

Comment 11: L65-78: What about the positive control and cut-off calculation. Please add such important information.

Response: We thank the reviewer for the valuable comments. The positive control and cut-off calculation have been added to the materials and methods on line 72-73 and 87-91, respectively.

            “Before the testing, positive serum controls were prepared by immunization of 50 healthy dairy cows with a formaldehyde-fixed P. multocida, as described previously [13].”

            “The cut-off value was 0.128 which was calculated from mean OD of negative control sera plus three standard deviations, as described in the previous study [4]. A sample was considered as a positive sample when both 1) the difference between the mean OD of the sample and with PBS alone, and 2) the difference between the mean OD of the sample and negative control ODs were greater than the cut-off value [4].”

Comment 12: L90: The disease was suggested if titers from 1:160 up to 1:1280 or higher were detected. The sample was considered positive at IHA test if….

Response: Thank you very much for your kind suggestion. The sentence is sound better, and it has been rephrased on line 103-104. (with a yellowing highlighted text).  

            “The samples were considered positive at IHA test if titers from 1:160 up to 1:1280 or higher were detected.”

Comment 13: L124-126: be more clear, you don´t need to inform the negative. From the 1,236 dairy cowas xxx (%) were positive to the ELISA and XXX (%) were positive to the IHA test.

Response: We thank the reviewer for your kind suggestion. The sentence has been rephrased on line 140-141. (with a yellowing highlighted text).  

            “The diagnostic test results are shown in Table 2. Of 1,236 dairy cows, 901 (73%) were positive to the ELISA test, and 860 (67%) were positive to the IHA test.”

Comment 14: L167: This finding suggests that more than half of Too vague, almost 2/3 would be also correct. Please rephrase

Response: Thank you very much for the kind suggestion. We agree with the reviewer. The sentence has been rephrased on line 178-180.

            “This finding suggests that almost 2/3 of the dairy cows in this area possessed antibodies against P. multocida and immunity may have been a result of the recent exposure to the pathogen.”  

Comment 15: L192: B:2 Sometimes (B:2). Be consistent.

Response: We thank the reviewer for the kind suggestion. We agree with the reviewer. The word has been replaced from “B:2” to “(B:2)” on line 188. (with a yellowing highlighted text).  

            “an indirect ELISA using P. multocida (B:2) outer membrane proteins as an antigen could range from”

Round 2

Reviewer 2 Report

The authors provided answers to all my comments and I am satisfied with the new information provided and with the extensive revision in the manuscript. I still believe that the manuscript has imitation but it can be published in the journal. 

There is some issue that still must be addressed regarding one of my comments: 

Comment 5: Since the OIE recommends the IHA as diagnostic test, I would like to see the Se and Sp results for the ELISA compared with IHA as gold standard. These results would improve your discussion.

Response: We thank the reviewer for the comments and providing us of the recent studies. This point is importance. We have analyzed according to the reviewer suggestions and found that the sensitivity of the ELISA test was 74%, and the specificity of the ELISA was 30% when compared with the IHA test as the gold standard.

The authors reply only in the reviewer's response. This information must be presented in the manuscript. The results from the classical epidemiological calculation and a discussion showing how the Bayesian approach is more adequate, comparing such results. This will strengthen the manuscript. 

The low Se and Sp of the IHA is the main problem, thus they solve this problem using a different approach that results in more reliable data since the ELISA was previously validated. This must be discussed in the manuscript. Comparing the results from the Bayesian approach to classical epidemiological calculations.  

Author Response

Authors Responses to the Reviewer(s)' Comments to Author:

Manuscript number: vetsci-922394

Manuscript title: Validation of a novel ELISA for the diagnosis of haemorrhagic septicaemia in dairy cattle from Thailand using a Bayesian approach.

Article type: Research Paper

Journal: Veterinary Sciences

Authors: Tawatchai Singhla, Pallop Tankaew, Nattawooti Sthitmatee

Response to reviewer 2 comments (Round 2):

There is some issue that still must be addressed regarding one of my comments: 

Comment 5: Since the OIE recommends the IHA as diagnostic test, I would like to see the Se and Sp results for the ELISA compared with IHA as gold standard. These results would improve your discussion.

Response: We thank the reviewer for the comments and providing us of the recent studies. This point is importance. We have analyzed according to the reviewer suggestions and found that the sensitivity of the ELISA test was 74%, and the specificity of the ELISA was 30% when compared with the IHA test as the gold standard.

The authors reply only in the reviewer's response. This information must be presented in the manuscript. The results from the classical epidemiological calculation and a discussion showing how the Bayesian approach is more adequate, comparing such results. This will strengthen the manuscript. 

The low Se and Sp of the IHA is the main problem, thus they solve this problem using a different approach that results in more reliable data since the ELISA was previously validated. This must be discussed in the manuscript. Comparing the results from the Bayesian approach to classical epidemiological calculations.  

Response: We thank the reviewer for the valuable suggestions. We absolutely agree with the reviewer.

The information of the classical epidemiological calculation has been added in the Materials and Methods and the Results, and this point has been discussed in the Discussion. These have been shown on line 138-140, 171-172 and 243-247 (with a yellowing highlighted text).  

on line 138-140;

“As an OIE-recommended assay for detection of antibodies against P. multocida, the IHA test was also used as the gold standard to estimate the Se and Sp of the ELISA test by classical epidemiological calculation or direct comparison.”

on line 171-172;

“The classical epidemiology calculation demonstrated that the Se and Sp of the ELISA test were 74% and 30%, respectively which lower than the results from the Bayesian approach.”

on line 243-247;

“Moreover, the Se and Sp of the ELISA test from the classical epidemiology calculation were lower than the estimation of the Bayesian analysis and quite difference from the previous validation [4]. This indicates that the Bayesian approach is the reliable method for estimation of the test performance when the reference test (the IHA test) is imperfect.”
